# THE DETECTION OF DISTRIBUTIONAL DISCREPANCY FOR TEXT GENERATION

## ABSTRACT

The text generated by neural language models is not as good as the real text. This means that their distributions are different. Generative Adversarial Nets (GAN) are used to alleviate it. However, some researchers argue that GAN variants do not work at all. When both sample quality (such as Bleu) and sample diversity (such as self-Bleu) are taken into account, the GAN variants even are worse than a well-adjusted language model. But, Bleu and self-Bleu can not precisely measure this distributional discrepancy. In fact, how to measure the distributional discrepancy between real text and generated text is still an open problem. In this paper, we theoretically propose two metric functions to measure the distributional difference between real text and generated text. Besides that, a method is put forward to estimate them. First, we evaluate language model with these two functions and find the difference is huge. Then, we try several methods to use the detected discrepancy signal to improve the generator. However the difference becomes even bigger than before. Experimenting on two existing language GANs, the distributional discrepancy between real text and generated text increases with more adversarial learning rounds. It demonstrates both of these language GANs fail.

## 1 INTRODUCTION

Text generation by neural language models (LM), such as LSTM (Hochreiter & Schmidhuber, 1997) have given rise to much progress and are now used to dialogue generation (Li et al., 2017), machine translation (Wu et al., 2016) and image caption (Xu et al., 2015). However, the generated sentences are still poor in semantics or global coherence, even not perfect grammatically speaking (Caccia et al., 2019).

It means that the discrepancy between generated text and real text is large. One reason is the architecture and parameters' number of LM itself (Radford et al., 2019; Santoro et al., 2018). Many researchers attribute it to the exposure bias (Bengio et al., 2015) because the LM is trained with a maximum likelihood estimate (MLE) and predicts the next word conditioned on words from the ground-truth during training. But it only conditions on the words generated by itself during reference.

Statistically, this discrepancy means the two distributional functions of real texts and generated texts is different. Reducing this distributional difference may be a practicable way to improve text generation.

Some researchers try to reduce this difference with GAN (Goodfellow et al., 2014). They use a discriminator to detect the discrepancy between real samples and generated samples, and feed the signal back to upgrade the generator (a LM). In order to solve the non-differential issue that arises by the need to handle discrete tokens, reinforcement learning (RL) (Williams, 1992) is adapted by SeqGAN (Yu et al., 2017), RankGAN (Lin et al., 2017), and LeakGAN (Guo et al., 2018). The Gumble-Softmax is also introduced by GSGAN (Jang et al., 2017) and RelGAN (Nie et al., 2019) to solve this issue. These language GANs pre-train both the generator ($G$) and the discriminator ($D$) before adversarial learning[1]. During adversarial learning, for each round, the $G$ is trained several epochs and then, the $D$ is trained tens of epochs. Learning stops when the model converges. Furthermore, considering the generated texts' quality and diversity simultaneously (Shi et al., 2018),

---

[1]An exception is RelGAN which needs not pre-train $D$.

MaskGAN (Fedus et al., 2018), DpGAN (Xu et al., 2018), FMGAN (Chen et al., 2018) and RelGAN (Nie et al., 2019) are proposed. They evaluate the generated text with Bleu and self-Bleu (Zhu et al., 2018) or LM socre and reverse LM score (Cífka et al., 2018), and claim these GANs improve the performance of generator.

However recently questions have been rasied over these claims. Semeniuta et al. (2018) and Caccia et al. (2019) showed that via more precise experiments and evaluation, these considered GAN variants are defeated by a well-adjusted language model . d'Autume et al. (2019) trained language GANs from scratch, nevertheless, they only achieve the "comparable" performance against LM. He et al. (2019) quantifies the exposure bias and concludes it is either 3 percent lower in performance or indistinguishable.

All the aforementioned methods treat GAN as a black box for evaluation. For those language GANs, there are several critical issues such as whether the $D$ detects the discrepancy or not; the detected discrepancy is severe or not, the signals from $D$ could improve the generator or not are still unclear. In this paper, we try to solve these problems via investigating GAN in both pre-training and the adversarial learning process. Theoretically analysing the signal from $D$, we obtain two metric functions to measure the distributional difference. With these two functions, we first measure the difference between the real text and the generated text by a MLE-trained language model (pre-train). Second, we try some methods to update generator with feedback signal from $D$, then, we use these metric functions to evaluate the updated generator. Finally, we analysis the existing language GANs during the adversarial learning with these two functions. All the code and data could be find https://github.com/.

Our contributions are as follows:

- We propose two metric functions to measure the distributional difference between real text and generated text. Besides that, a method is put forward to estimate them.
- Evaluated using these two functions, a number of experiment show there is an obvious discrepancy between the real text and the generated text even when it is generated by a well-adjusted language model.
- Although this discrepancy could be detected by $D$, the feedback signal from $D$ can not improve $G$ using existing methods.
- Experimenting on two existing language GANs, SeqGAN and RelGAN, the distributional discrepancy between real text and generated text increases with more adversarial learning rounds. It demonstrates both of these language GANs fail.

## 2 METHOD

In GAN, the generator $G_\theta$ implicitly defines a probability distribution $p_\theta(x)$ to mimic the real data distribution $p_d(x)$.

$$\min_{G_\theta} \max_{D_\phi} V(D_\phi, G_\theta) = \mathbb{E}_{x \sim p_d(x)}\left[log D_\phi(x)\right] + \mathbb{E}_{x \sim p_\theta(x)}\left[log\big(1 - D_\phi(x)\big)\right] \quad (1)$$

We define $D_\phi$ to detect the discrepancy between $p_\theta(x)$ and $p_d(x)$. We optimize $D_\phi$ as follow,

$$\max_{D_\phi} V(D_\phi, G_\theta) = \max_{D_\phi} \mathbb{E}_{x \sim p_d}\left[log D_\phi(x)\right] + \mathbb{E}_{x \sim p_\theta}\left[log\big(1 - D_\phi(x)\big)\right] \quad (2)$$

Assuming $D_\phi^*(x)$ is the optimal solution for a given $\theta$, according to (Goodfellow et al., 2014), there will be,

$$D_\phi^*(x) = \frac{p_d(x)}{p_d(x) + p_\theta(x)} \quad (3)$$

We obtain two metric functions to measure this discrepancy.

$$\begin{cases} D_\phi^*(x) \geq 0.5, & iif \quad p_d(x) \geq p_\theta(x) \\ D_\phi^*(x) < 0.5, & iif \quad p_d(x) < p_\theta(x) \end{cases} \tag{4}$$

With it, the integration of density function could be transformed into statistic equation. Based on that, we could get a way which will be described in next, to compute the precise discrepancy.

## 2.1 APPROXIMATE DISCREPANCY

Let,

$$q_d(x) = \frac{p_d(x)}{p_d(x) + p_\theta(x)} \qquad q_\theta(x) = \frac{p_\theta(x)}{p_d(x) + p_\theta(x)} \tag{5}$$

So, $q_d(x) = p(x$ comes from real data$|x)$, $q_\theta(x) = p(x$ comes from generated data$|x)$, $q_d(x) + q_\theta(x) = 1$. With equation 5, we could get a constraint and an approximated measure function of distributional function. Figure 1(a) illustrates the relationship between $q_\theta(x)$ and $q_d(x)$.

Let,

$$u_d = \mathbb{E}_{x \sim p_d(x)}\big(D_\phi^*(x)\big) \qquad u_\theta = \mathbb{E}_{x \sim p_\theta(x)}\big(D_\phi^*(x)\big) \tag{6}$$

They are two statistic equations who are the expectation of the $D_\phi^*$'s predictions on real text and on generated text respectively. According to the above equation, it is easy to get following equation.

$$\frac{1}{2}\big[u_d + u_\theta\big] = 0.5 \tag{7}$$

It gives a constraint for $D_\phi$ converging to $D_\phi^*$. We should take this constraint into account when estimating the ideal function $D_\phi^*$. From equation 3, the process of optimizing the discriminator is make $u_d$ big and make $u_\theta$ small. So, we could estimate the distributional discrepancy according to the following function.

Intuitively, using $u_d$ and $u_\theta$, we get a metric function to measure the discrepancy between $p_\theta(x)$ and $p_d(x)$,

$$d_a = u_d - u_\theta \tag{8}$$

We call it approximate discrepancy. It is the subtraction of the average score of a well-trained discriminator (denoted as $\hat{D}_\phi$) makes the predictions on real samples and on generated samples. It reflects the discrepancy between these two sets to some degree. From equation 5 and 6, we get equation 8,

$$d_a = \int \big[q_d(x) - q_\theta(x)\big]p_d(x)dx = \mathbb{E}_{x \sim p_d(x)}\big[q_d(x) - q_\theta(x)\big] \tag{9}$$

Figure 1 (a) illustrates the discrepancy between two distributional functions $q_\theta(x)$ and $q_d(x)$. Both of them are systematic to the line of $q = 0.5$. But it is not a complete measure because there is not only a positive part but also a negative part. A complete metric function is shown in next section.

## 2.2 ABSOLUTE DISCREPANCY

In order to precisely measure the discrepancy, we define $d_s$,

$$d_s = \frac{1}{2}\int \big|p_d(x) - p_\theta(x)\big|dx \tag{10}$$

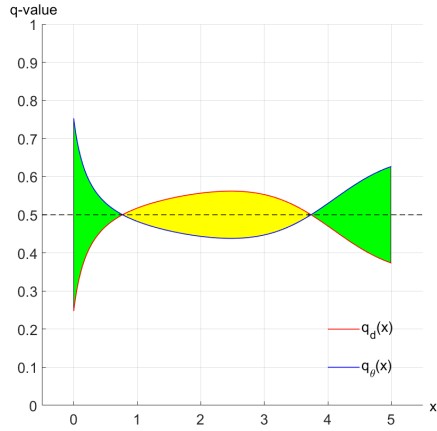

(a) Approximate discrepancy illustration.

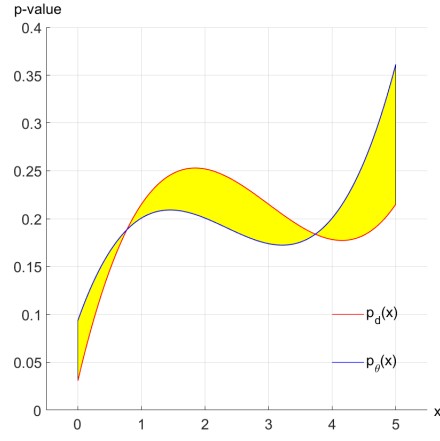

(b) Precise discrepancy illustration.

Figure 1: The illustration of two measures. In (a), the yellow area denotes the negative and the green one denotes the positive. In (b), the half of the shadow area equals to the result of equation 11. Therefore, the range of it is $0 \sim 2$. Bigger value means more discrepancy.

The range of this function is $0 \sim 1$. The bigger of its value is, the discrepancy is more. When its value is zero, it means $p_d(x) \equiv p_\theta(x)$, namely there is no discrepancy. Fortunately, this function could be estimated by statistic method which is described by following equation. The proof is shown in appendix A.

$$d_s = \frac{1}{2}\left[ \mathop{\mathbb{E}}_{\substack{x \sim p_d(x) \\ D_\phi^*(x) > 0.5}} \left(1\right) - \mathop{\mathbb{E}}_{\substack{x \sim p_d(x) \\ D_\phi^*(x) \leq 0.5}} \left(1\right) + \mathop{\mathbb{E}}_{\substack{x \sim p_\theta(x) \\ D_\phi^*(x) \leq 0.5}} \left(1\right) - \mathop{\mathbb{E}}_{\substack{x \sim p_\theta(x) \\ D_\phi^*(x) > 0.5}} \left(1\right) \right] \qquad (11)$$

With equation 11, we could more precisely estimate the discrepancy between $p_\theta(x)$ and $p_d(x)$. Assuming the classification precision of $D_\phi^*$ is $a$, then the error rate is $b = 1 - a$. According to equation 11, $d_s = a - b$. So, the discrepancy between $p_\theta(x)$ and $p_d(x)$ equals the classification precision of $D_\phi^*$ minus its error rate.

## 2.3 USING $D_\phi^*(x)$ TO IMPROVE $G_\theta$

Given an instance $x$ generated by $G_\theta$, if $D_\phi^*(x)$ is larger, it means the possibility of $x$ is real data is larger. For an instance $D_\phi^*(x) = 0.8$, there will be $p_\theta(x) < p_d(x)$ according to equation 3. So, we should update $G_\theta$ to make the probability density $p_\theta(x)$ increase. It may improve the performance of $G_\theta$. Based on this, we could select out some generated instances by the value of $D_\phi^*(x)$ to update the generator. In fact, we find it helpful to use the faked samples whose score are higher assigned by $D_\phi^*$. Experiment 4.3 shows the results.

## 3 IMPLEMENT PROCEDURE

The optimal function $D_\phi^*$ is an ideal function which can only be statistically estimated by approximated function. We could design a function $D_\phi$ and sample from real data and generated data, then train $D_\phi$ according to equation 2. When it convergences, we get $\hat{D}_\phi$. $\hat{D}_\phi$ is the approximated function of $D_\phi^*$. The degree of approximation is mainly determined by three factors: the structure and the parameters' number of $D_\phi$, the volume of training data, and the settings of hyper-parameters.

Based on the above analysis, we get two metric functions to measure the distributional discrepancy between dataset $A$ and $B$. The specific implement procedure is as follow,

Step 1: Design a discriminator $D_\phi$.

Step 2: The set $A$ and $B$ are respectively divided into training set $D_{trainA}$ and $D_{trainB}$, validation set $D_{devA}$ and $D_{devB}$, and test set $D_{testA}$ and $D_{testB}$. The partition should be as equal an amount of instances as possible for classification training.

Step3: $D_\phi$ is optimized with $D_{trainA}$ and $D_{trainB}$ according to the equation 2. Validated with $D_{devA}$ and $D_{devB}$, we could judge whether $D_\phi$ convergences or not and then get $\hat{D}_\phi$.

Step4: According to equation 8 and 11, with two test datasets, we could estimate the discrepancy of two distributional density functions between dataset $A$ and $B$. $\hat{d}_s$ denotes the absolute discrepancy and $\hat{d}_a$ denotes the approximate discrepancy respectively.

Generally speaking, there should be $d_s \leq \hat{d}_s$. Because $D_\phi^*$ can not be obtained, it is hard to get the degree of the approximation $d_s$ to $\hat{d}_s$. Many research results have shown that the discriminators with deep neural networks are very powerful, and can even exceed human performance on some tasks such as image classification (He et al., 2016) and text classification (Kim, 2014). So, if the $D_\phi$ with CNN and attention mechanism is well trained, $\hat{D}_\phi$ will be a meaningful approximation of $D_\phi^*$. Therefore, we could obtain the meaningful approximation of $d_s$ and $d_a$ via $\hat{D}_\phi$.

## 4 EXPERIMENT

We select out SeqGAN and RelGAN as representatives for experiment and the benchmark datasets are also the same as theirs. Then, we show that the well trained discriminator $D$ could measure the discrepancy between the real and generated texts, and then point out the existing GAN-based methods do not work. Finally, a third party discriminator is used to evaluate the performance of adversarial learning with the increment of training iterations.

### 4.1 DATASETS AND MODEL SETTINGS

Both SeqGAN and RelGAN are experimented on relative short sentences (COCO image caption) [2] and long sentences dataset (EMNLP2017 WMT news) [3]. For the former dataset, sentences average length is about 11 words. There are total 4,682 word types and the longest sentence consists of 37 words. Both the training and test data contain 10,000 sentences. For the latter, the average length of sentences is about 20 words. There are total 5,255 word types and the longest sentence is consisted of 51 words. All training data, about 280 thousand sentences, is used and 10,000 sentences in test data. According to section 3, each test data divide into two parts. Half is validation set and the rest half is test set. We always generate the same amount of sentences to compare with the two test datasets respectively.

For these two models, all hyper-parameters including word embedding size, learning rate and dropout are set the same as their papers. For RelGAN, the standard GAN loss function (the non-saturating version) is adapted because the relative standard loss which is used in (Nie et al., 2019) does not meet the constraints of equation 7. But, when measure RelGAN's discrepancy during the adversarial stage, it own loss function is still relative standard loss. A critical hyper-parameters, temperature, is set 100 which is the best result in their paper. During the process of training $D$ to obtain, we always train $D$ 10,000 epochs and observe its performance on validation dataset.

### 4.2 THE DISTRIBUTIONAL DIFFERENCE IN PRE-TRAIN

We estimate the distributional difference caused by the MLE-based generators. We first train the generator for $N$ epochs and then train $D$ until it converges (it is trained 10 thousand epochs). For example, following (Nie et al., 2019), we train $G$ 150 epochs, and at that time, the PPL is the smallest value measured by validation set. Then, $D_\phi$ is trained following the procedure in section 3. Figure 2 shows the results on EMNLP dataset. From this figure, we can see that (1) $D_\phi$ convergences after

---

[2]http://cocodataset.org/
[3]http://www.statmt.org/wmt17/

about 5,000 epochs. (2) There is always $u_d + u_\theta \approx 1$ everywhere. (3) When it convergence, the $u_d$ is 0.7 and $u_\theta$ is 0.3. More results are illustrated in appendix B.1.

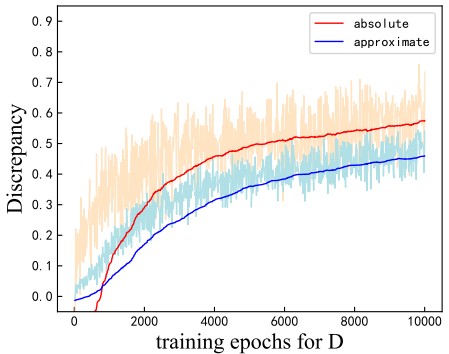 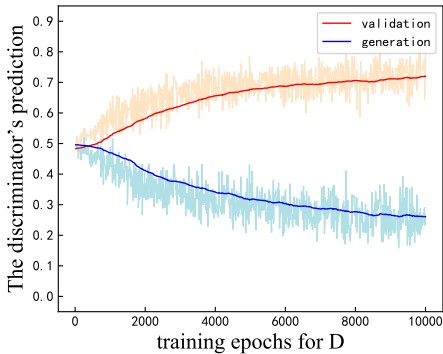

(a) The discrepancy between validation set and generated data. The orange lines denote the absolute discrepancy and the blue lines denote the approximate discrepancy.

(b) The discriminator's prediction. The orange lines denote the predictions on validation set and the blue lines denote on generated data.

Figure 2: The results of pre-train SeqGAN' generator 80 epochs on EMNLP dataset. All the pale lines denote batch instances' discrepancy and the curve is the exponential moving average on this sampled batch for each epoch.

Considering the smoothed value on one batch rather than the prediction on the whole data, we use the convergence discriminator to predict on the all validation data and generated data [4]. Table x summaries the discrepancy across two models and two datasets. It reflects the difference between real text and generated text is huge.

Table 1: The discrepancy across two models and two datasets in pre-train. For both $d_a$ and $d_s$, lower is better.

| Model | Dataset | $\hat{d}_s$ | $\hat{d}_a$ | Accuracy |
|---|---|---|---|---|
| SeqGAN | COCO | 0.42 | 0.44 | 0.71 |
| | EMNLP | 0.57 | 0.47 | 0.78 |
| RelGAN | COCO | 0.64 | 0.45 | 0.82 |
| | EMNLP | 0.52 | 0.31 | 0.76 |

### 4.3 COULD THE DETECTED DISCREPANCY BY $\hat{D}_\phi$ IMPROVE THE GENERATOR?

In this section, we will explore whether the above discrepancy detected by $\hat{D}_\phi$ in pre-train could improve $G$. It should be noted that the $\hat{D}_\phi$ is well pre-trained. We select out the best pre-train epochs for $G$. It is updated according to the signals from the $\hat{D}_\phi$. In order to verify the effect of those feedback signals, we generate many instances rather only several batch-size ones are used to adjust generator's parameters.

Then, fixing $G$, we re-train $D$ with 10,000 epochs to get a new convergence discriminator, named $\hat{D}'_\phi$, for computing two distributional functions according to equation 9 and 11. Unfortunately, in the view of absolute discrepancy or approximated discrepancy, the discrepancy always overpass the original value which is computed in pre-train. This demonstrates that the generator is not improved yet. Figure 3 illustrates a comparison. More experimental results are shown in appendix B.2.

Besides following (Zhu et al., 2018), We propose a new methods train $G$. Rather than all the generated instances are used to update $G$, only the ones who are assigned relative low scores by $D$

---

[4]According to section 3, we sample generated instances as much as test instances.

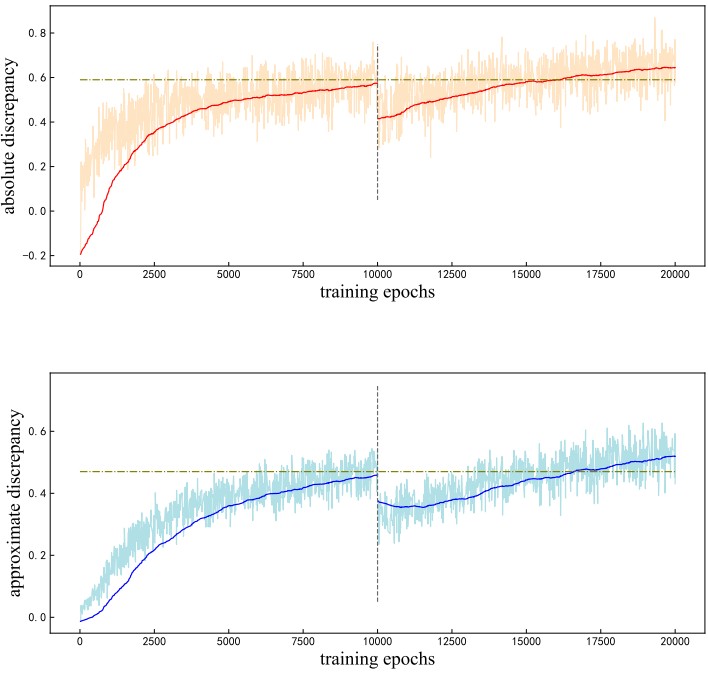

Figure 3: The compare of discrepancy between pre-train and the generator is updated with the feedback signals from $\hat{D}_\phi$ which is obtained from pre-train. The vertical dash line represents the end of pre-training. SeqGAN's generator is pre-trained 80 epochs and the dataset is COCO.

are used. We denote it as HW. The reason is that we think the higher score instances maybe more informative than the lower ones. Regretfully, both of two methods fail. Table 3 list the discrepancy across two datasets. It again demonstrated that the absolute measure is necessary. Appendix B.3 show the results of several thresholds are set for selecting out more informative generated instances.

Table 2: The compare between the absolute discrepancy in pre-training and $G$ is updated by $\hat{D}_\phi$'s feedback signal. #samples denotes the amount of the generated data is used for updating the $G$. For example, 2S means the generated instances is two time as the amount of test data. Random denotes the existing way but the other row are the results according to HW. $< 0.3 - 0.5$ means the generated instances whose score are between 0.3 and 0.5 assigned by $D$, are selected out.

| Dataset | COCO | | | | | EMNLP | | | | |
|---|---|---|---|---|---|---|---|---|---|---|
| pre-Train | **0.42** | | | | | **0.57** | | | | |
| #samples | 0.1S | 0.5S | 1S | 2S | 5S | 0.1S | 0.5S | 1S | 2S | 5S |
| random | 0.52 | 0.45 | 0.54 | 0.53 | 0.55 | 0.68 | 0.68 | 0.67 | 0.67 | 0.67 |
| $< 0.3$ | 0.60 | 0.58 | 0.54 | 0.54 | 0.72 | 0.77 | 0.76 | 0.73 | 0.76 | 0.73 |
| $0.3 - 0.5$ | 0.44 | 0.56 | 0.58 | 0.60 | 0.55 | 0.68 | 0.67 | 0.66 | 0.63 | 0.66 |
| $0.5 - 0.9$ | 0.52 | 0.51 | 0.47 | 0.54 | 0.58 | 0.66 | 0.57 | 0.60 | 0.64 | 0.62 |
| $\geq 0.9$ | 0.68 | 0.46 | 0.50 | 0.44 | 0.58 | 0.66 | 0.60 | 0.59 | 0.60 | 0.62 |

## 4.4 A THIRD PARTY DISCRIMINATOR EVALUATES THESE LANGUAGE GANS

In order to evaluate different adversarial learning's GANs, we use a third party discriminator $D_3$ which is a clone of the discriminator in its counterpart language GAN except parameters' value. Given a round, we train $D_3$ many epochs (making sure it convergence) with real text and the generated text at this round. Then, two distributional functions are computed according to its prediction. Figure 4 shows the result. In the view of both approximate discrepancy and absolute discrepancy,

the difference of the distribution on real text and generated text does not decrease with more adversarial learning rounds are adapted. Once again, it shows that the way of the existing language GANs could not improve text generation.

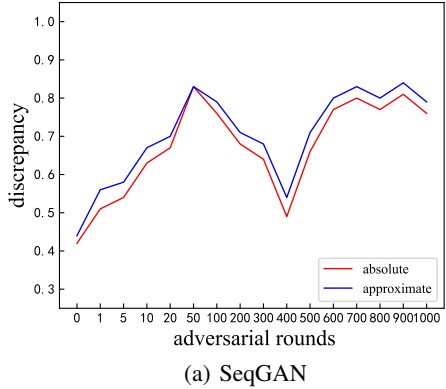

(a) SeqGAN

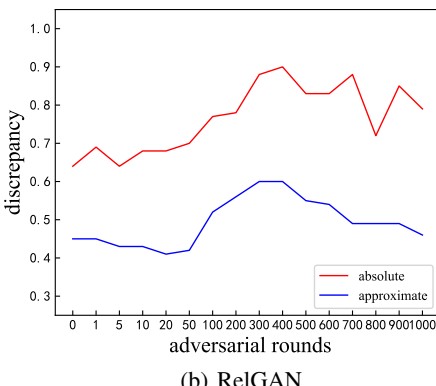

(b) RelGAN

Figure 4: A third party discriminator evaluates two GANs' performance variety on COCO.

## 5 RELATED WORK

Many GAN-based models are proposed to improve traditional neural language model. SeqGAN (Yu et al., 2017) is the first one and try by to attack the non-differential issue by resorting RL. By applying policy gradient (Sutton et al., 2000) method, it optimizes the LSTM generator with rewards received through Monte Carlo (MC) sampling. Many researchers, such as RankGAN (Lin et al., 2017), MailGAN (Tong et al., 2017) follow this way although its in-effective in MC search. The RL-free model, for an example GSGAN (Jang et al., 2017), contains applying continuous approximating softmax function and working on latent continuous space directly. TextGAN (Salimans et al., 2016) adds Maximum Mean Discrepancy to the original objective of GAN based on feature matching. RelGAN (Nie et al., 2019) is a state-of-the-art model which uses relation memory (Santoro et al., 2018), which allowing for interactions between memory slots by using the self-attention mechanism (Vaswani et al., 2017). We select out SeqGAN and RelGAN as representatives for experiment. The results show that the adversarial learning does not work in both of them.

Caccia et al. (2019) first argues the current evaluation measures correlate with human judgment (Cífka et al., 2018) is treacherous. They furthermore propose temperature sweep which evaluates model at many temperature settings rather than only one. By using this metric, they find a well-adjust language model could beat those considered language GANs. Semeniuta et al. (2018) and He et al. (2019) also argues GAN-based models are weak than LM, because they observe the impact of exposure bias is not severe. He et al. (2019) furtherly quantify the exposure bias by using conditional distribution. Neither designing a better metric nor showing the weakness of the language GANs, we try to investigate language GANs in mechanism and quantify the discrepancy between real texts and generated texts both in pre-train and adversarial learning.

## 6 CONCLUSION AND FUTURE WORK

We present two directly metric functions to measure the discrepancy between real text and generated text. It must be noted that they are independent of any text generation method including GANs-based. Numerous experiments show that this discrepancy dose exist. We try some methods to update the parameters of generator according to the detected discrepancy signals. Unfortunately, the distributional difference between real data and generated data does not decrease. It is hard to improve generator with these signals. Finally, We use a third part discriminator to evaluate the effectiveness of GAN and find with more adversarial learning epochs, the discrepancy increase rather than decreasing. It shows the existing language GANs do not work at all.

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

## A   FORMULA INDUCTION

We show the derivation of Equation 11,

$$
\begin{aligned}
d_s &= \frac{1}{2} \int \big| p_d(x) - p_\theta(x) \big| dx \\
&= \frac{1}{2} \left[ \int_{p_d(x) \geq p_\theta(x)} \big( p_d(x) - p_\theta(x) \big) dx + \int_{p_d(x) < p_\theta(x)} \big( p_\theta(x) - p_d(x) \big) dx \right] \\
&= \frac{1}{2} \left[ \int_{p_d(x) \geq p_\theta(x)} p_d(x) dx + \int_{p_d(x) < p_\theta(x)} p_\theta(x) dx - \int_{p_d(x) \geq p_\theta(x)} p_\theta(x) dx - \int_{p_d(x) < p_\theta(x)} p_d(x) dx \right] \\
&= \frac{1}{2} \left[ \mathbb{E}_{\substack{x \sim p_d(x) \\ p_d(x) \geq p_\theta(x)}} \big( 1 \big) + \mathbb{E}_{\substack{x \sim p_\theta(x) \\ p_d(x) < p_\theta(x)}} \big( 1 \big) - \mathbb{E}_{\substack{x \sim p_\theta(x) \\ p_d(x) \geq p_\theta(x)}} \big( 1 \big) - \mathbb{E}_{\substack{x \sim p_d(x) \\ p_d(x) < p_\theta(x)}} \big( 1 \big) \right] \\
&= \frac{1}{2} \left[ \mathbb{E}_{\substack{x \sim p_d(x) \\ z \geq 0.5}} \big( 1 \big) + \mathbb{E}_{\substack{x \sim p_\theta(x) \\ z < 0.5}} \big( 1 \big) - \mathbb{E}_{\substack{x \sim p_\theta(x) \\ z \geq 0.5}} \big( 1 \big) - \mathbb{E}_{\substack{x \sim p_d(x) \\ z < 0.5}} \big( 1 \big) \right]
\end{aligned}
\tag{12}
$$

where $z = \frac{p_d(x)}{p_d(x) + p_\theta(x)}$.

# B APPENDIX B

## B.1 THE REST THREE EXAMPLES IN PRE-TRAIN.

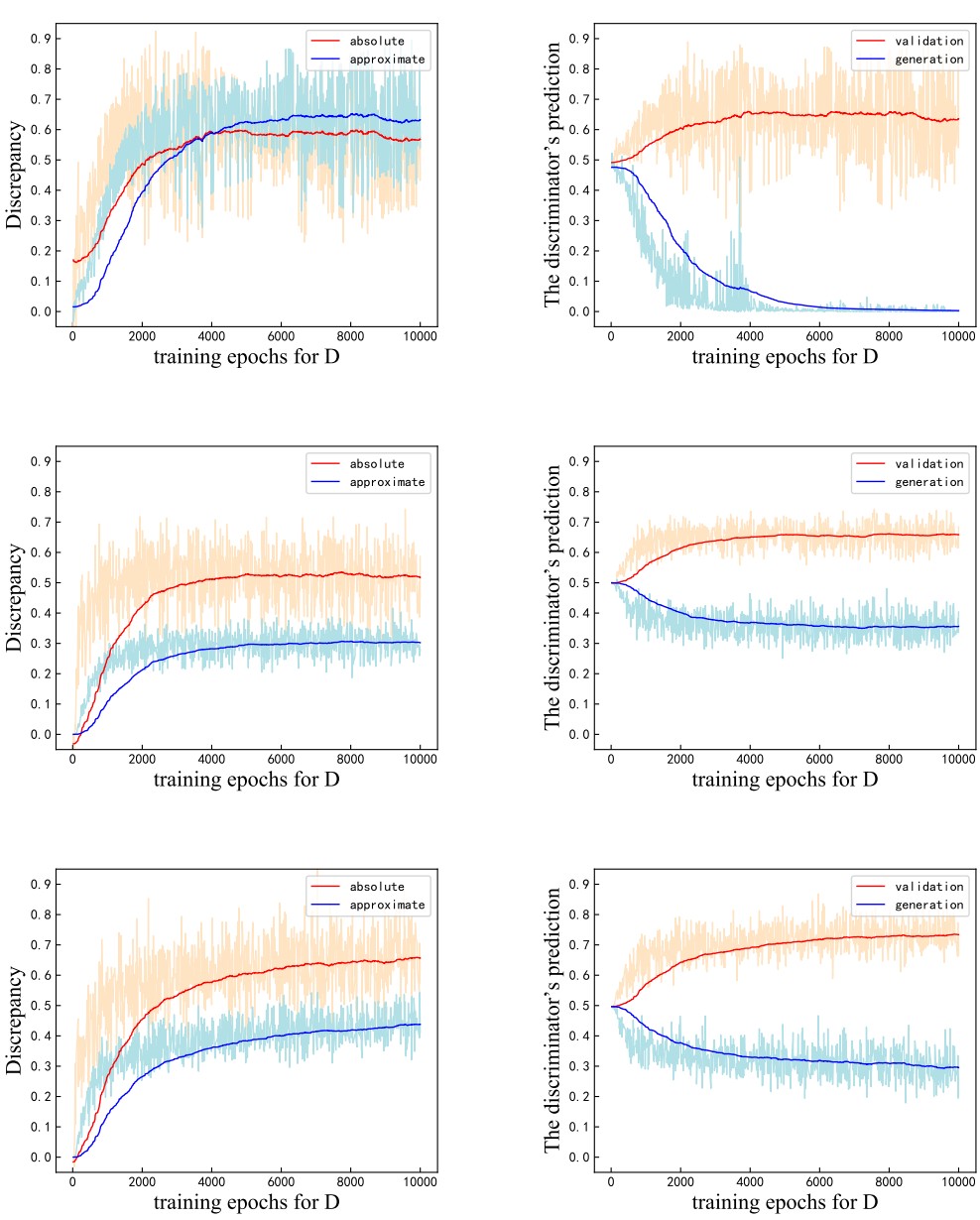

Figure 5: The results of pre-train SeqGAN' generator 80 epochs on COCO dataset (upper), RelGAN on EMNLP (middle) and RelGAN on COCO(below). All the pale lines denote batch instances' discrepancy and the curve is the exponential moving average on this sampled batch for each epoch.

## B.2 THE DISCREPANCY COMPARE ON EMNLP BETWEEN PRE-TRAINING AND UPDATED LM.

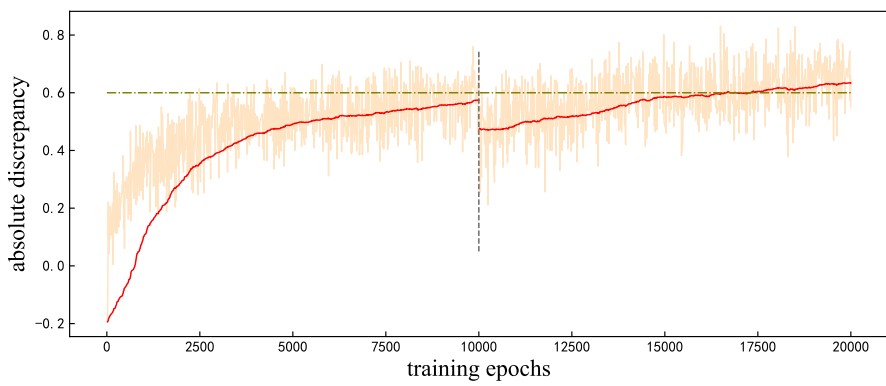

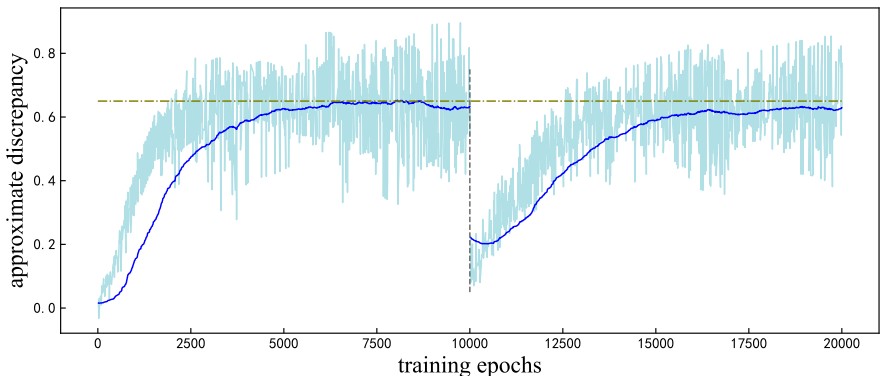

Figure 6: The compare of discrepancy between pre-train and the generator is updated with the feedback signals from $\hat{D}_\phi$ which is obtained from pre-train. The vertical dash line represents the end of pre-training. SeqGAN' generator is pre-trained 80 epochs and the dataset is EMNLP.

## B.3 THE COMPARE OF THE APPROXIMATE DISCREPANCY BETWEEN RANDOM AND HW.

Table 3: The compare between the approximate discrepancy in pre-training and $G$ is updated by $\hat{D}_\phi$'s feedback signal. #samples denotes the amount of the generated data is used for updating the $G$. For example, 2S means the generated instances is two time as the amount of test data. Random denotes the existing way but the other row are the results according to HW. $< 0.3 - 0.5$ means the generated instances whose score are between 0.3 and 0.5 assigned by $D$, are selected out.

| Dataset | COCO | | | | | EMNLP | | | | |
|---|---|---|---|---|---|---|---|---|---|---|
| pre-Train | **0.44** | | | | | 0.47 | | | | |
| #samples | 0.1S | 0.5S | 1S | 2S | 5S | 0.1S | 0.5S | 1S | 2S | 5S |
| random | 0.57 | 0.50 | 0.58 | 0.58 | 0.60 | 0.55 | 0.52 | 0.50 | 0.52 | 0.53 |
| $< 0.3$ | 0.65 | 0.62 | 0.59 | 0.60 | 0.75 | 0.64 | 0.62 | 0.61 | 0.61 | 0.61 |
| $0.3 - 0.5$ | 0.49 | 0.60 | 0.62 | 0.64 | 0.56 | 0.51 | 0.50 | 0.49 | 0.49 | 0.49 |
| $0.5 - 0.9$ | 0.57 | 0.55 | 0.51 | 0.58 | 0.62 | 0.49 | 0.46 | 0.45 | 0.45 | **0.44** |
| $\geq 0.9$ | 0.55 | 0.55 | 0.50 | 0.61 | 0.47 | 0.47 | 0.45 | 0.45 | 0.45 | 0.46 |

## C  GENERATED SENTENCES ON COCO IMAGE CAPTIONS DATASET

Table 4: Generated sentences who are scored 0.9 or higher by $\hat{D}_\phi$ at the end of pre-training. Obviously, they are better the next sentences listed in table 5
.

| |
|---|
| the ground and sink are under the window of a pink covered pot . |
| a toilet sitting next to a toilet next to a green organized kitchen . |
| the bathroom contains a mop that rolls of flowers next to the vanity do two sinks . |
| a vehicle trailer meeting a corner of a street . |
| a small dog is sitting in a purse with text . |
| two men are riding an outside of the window of a street |
| a porcelain tea pot by a window |
| a humongous jumbo jet with chain chair on the ground . |
| large white bus some luggage onto one the other smiling no front tire . |
| a wooden ship with rusted heater flying high in a sky . |
| the side outside with a giraffe doors |
| the bathroom has a pedestal table with pots on it |
| bathroom with wooden cabinets and a washer , curtain and bottles growing on the flooring . |
| the side of a person looks at a woman on the back seat . |
| a bottle of whiskey in the bathroom with broken toilet |
| a ramp extends to the top of an air mattress . |
| a photograph of a palm wall in the glass 's lap top . |
| a smiley face stand with 4 per gallon . |
| several bicycles parked stuffed in a display using it . |
| two dogs on top of a car at an amusement table . |
| a white airplane flying above someone with his friend parked behind it terminal . |
| a bathroom water from a shower with cobbler and pink tile walls . |
| a toilet is bathroom with multiple monitors paper . |
| a red fire hydrant displaying the woman in a park and many headlights . |
| a tiled bathroom with a claw view of toilet and soap dispenser |
| a group of ripe bananas in their hands on the back seat . |
| a motorcycle parked next to a stone building next to the road . |
| an airplane flies low to people aboard a boat with a cemetery a frisbee . |
| a bath room with a toilet behind it |
| a homemade cake in home bathroom has a sink , a , shower , and a window . |
| a truck is shown as the parking lot corner . |
| a bathroom is lit with foods such as : to a web job graze . |
| porcelain toilet in a blue toilet seat . |
| a simple white bathroom with a large white fridge , and shower . |
| a large delta passenger airplane flying through a cloudy sky . |
| a bathroom with a mirror , dishwasher , dishwasher , and tub/shower . |
| a blue plane is walking as seen all towards the usual number of tonic . |
| two adorable chubby dogs in a group of pots |
| four airplanes sitting on top of a building in the blue sky . |
| horse is racing on his motorcycle while maintenance are walking next to the river . |
| an image of a old bathroom with a bottle of spirits suggest tuscan decor , laptop . |
| an outdoor art bus is facing toward another snowboarder |
| a kitchen filled with in tables and two computer monitors . |
| this is an image of a motorcycle parked down by a bench . |
| a crowd of people sitting on a bench next to street |
| an electronic cat in a mens restroom next to multiple dark and pans along under a screen table . |
| two woman in her phone next to a vehicle . |
| a person riding a motor moped at a playground . |
| two dirty dog standing in front of a car . |
| a white bathtub sits next to a mirror in a small closed . |

Table 5: Generated sentences who are scored 0.1 or lower by $\hat{D}_\phi$ at the end of pre-training. Obviously, they are better the next sentences listed in table.

---

a white metal structure with a backpack on top .
a bath tub next to the toilet in middle of a nice plate .
rams lights and need onto the outside of a zoo .
an airplane parked on the ground in front of a building .
a white vw car passing in front of a car .
an airplane flying over an airport terminal .
two people riding bikes on an outdoor from police officer .
an old propeller airplane parked off .
a man standing down by side on a street .
a image of a toilet , mirror and painting on the wall
two toilets walking a lipstick , a television .
two doves sit on a bathroom counter with wooden cabinets and a bucket .
this enclosure has a white toilet next to the sink .
the dining area is wide two people riding her back to carve above them .
two men stand with all line of street using indoors .
a boy wearing travel and his motorcycle outside a town outside .
a person brushing her teeth with her hair holding a stove .
a white bathtub sitting in a bathroom with no cabinets .
a bath room with two counter preparing food .
a man holding up a market stand by the ground .
a kitchen has large round glass serving and cabinets .
a woman standing outside of a black car .
a bathroom has an island in the middle granite glass .
a large passenger jet flying over an airport next to the street .
a tiger cat sitting on top of a window looking very clean .
bicyclists holding a flip phone next to the aircraft .
a walk opened in the bathroom with a jungle theme .
the sink has white appliances with a child .
a kitchen with a chrome toilet next to bathtub .
man standing in a blue park looking sidewalk next to a brown horse against a group of electrical boxes .
an old parked motorcycle with its kickstand down to .
a messy bathroom with a tub , in the sink and the mirror with no privacy .
a an open kitchen tucked in a public kitchen .
a white brown kitchen with bowls on a counter top counter .
a woman standing in red liquid under a small water fountain .
a view above the toilet roll a sink with a stove .
a couple of chefs up in a kitchen preparing food .
a bathroom with a toilet , and mirrors from its reflection in a large sink .
a bath room with a tub and refrigerator .
a group of traffic light with people skiing in the mist
two dogs are huddled together a screen corner on a wall .
wild animals grazing in the center of a blue sky .
a woman holds a spoon by off a road .
a kitchen with toiletries in the just darts .
motor sandwich and ride to turn across a road .
a man crossing a traffic in an oriental city
women turning her phone food prepares food .
two stuffed animals are laptop , dishwasher , and a sink .
a toilet in front of a window in a white room .
large woman , on snowboards at an open purse
an image of a sink , trash can .

---

