# OpenReview forum: "The Detection of Distributional Discrepancy for Text Generation"
_ICLR.cc/2020/Conference — Reject_

### Official Review · AnonReviewer3 · 2019-10-15
**Official Blind Review #3**

**Rating:** 3

**Review:**

This paper proposes an estimator to quantify the difference in distributions between real and generated text based on a classifier that discriminates between real vs generated text.  The methodology is however not particularly well motivated and the experiments do not convince me that this proposed measure is superior to other reasonable choices.  Overall, the writing also contains many grammatical errors and confusing at places.

Major Comments:

- There are tons of other existing measures of distributional discrepancy that could be applied to this same problem.  Some would be classical approaches (eg. Kullback-Leibler or other f-divergence based on estimated densities, Maximum Mean Discrepancy based on a specific text kernel, etc) while others would be highly related to this work through their use of a classifier.  Here's just a few examples:

i) Lopez-Paz & Oquab (2018). "Revisiting Classifier Two-Sample Tests
": https://arxiv.org/abs/1610.06545
ii) the Wasserstein critic in Wasserstein-GAN
iii) Sugiyama et al (2012). "Density Ratio Estimation in Machine Learning"

Given all these existing methods (I am sure there are many more), it is unclear to me why the estimator proposed in this paper should be better. The authors need to clarify this both intuitively and empirically via comparison experiments (theoretical comparisons would be nice to see as well).

- The authors are proposing a measure of discrepancy, which is essentially useful as a two-sample statistical test.  As such, the authors should demonstrate a power analysis of their test to detect differences between real vs generated text and show this new test is better than tests based on existing discrepancy measures.

- The authors claim training a generator to minimize their proposed divergence is superior to a standard language GAN. However, the method to achieve this is quite convoluted, and straightforward generator training to minimize D_phi does not appear to work (the authors do not say why either).


Minor Comments:

- x needs to be defined before equation (1).

- It is mathematically incorrect to talk about probability density functions when dealing with discrete text. Rather these should be referred to as probability mass functions, likelihoods, or distributions (not "distributional function" either).



**Experience Assessment:**

I have published in this field for several years.

**Review Assessment: Checking Correctness Of Derivations And Theory:**

I carefully checked the derivations and theory.

**Review Assessment: Checking Correctness Of Experiments:**

I carefully checked the experiments.

**Review Assessment: Thoroughness In Paper Reading:**

I read the paper thoroughly.

---

### Official Review · AnonReviewer2 · 2019-10-19
**Official Blind Review #2**

**Rating:** 1

**Review:**

This paper proposes two metrics to measure the discrepancy between generated text and real text, based on the discriminator score in GANs. Empirically, it shows that text generated by current text generation methods is still far from human-generated text, as measured by the proposed metric. The writing is a bit rough so sometimes it's hard to figure out what has been done. It's also unclear how the proposed metrics compare to simply using the discriminator for evaluation. Therefore, I'm inclined to reject the current submission.

Approach:
- The proposed metric essentially relies on the learned discriminator to measure the closeness of generated text vs real text, based on the strong assumption that the learned discriminator is near-optimal. It has been previously shown that learning a classifier from generated and real text does not generalize well (Lowe et al, 2017, Chaganty et al, 2018).
- What's the advantage of the proposed metric, compared to existing ones, e.g. KL divergence, total variation etc.?

Experiments:
- What's the accuracy of the learned discriminators? The discrepancy could be due to both data difference and classification error.

Minor:
Bleu -> BLEU

Reference:
Towards an automatic turing test: Learning to evaluate dialogue responses. R. Lowe, M. Noseworthy, I. V. Serban, N. Angelard- Gontier, Y. Bengio, and J. Pineau. 2017.
The price of debiasing automatic metrics in natural language evaluation. A. Chaganty, S. Mussmann, and P. Liang. 2018.

**Experience Assessment:**

I have published one or two papers in this area.

**Review Assessment: Checking Correctness Of Derivations And Theory:**

I carefully checked the derivations and theory.

**Review Assessment: Checking Correctness Of Experiments:**

I carefully checked the experiments.

**Review Assessment: Thoroughness In Paper Reading:**

I read the paper thoroughly.

---

### Official Review · AnonReviewer1 · 2019-10-23
**Official Blind Review #1**

**Rating:** 1

**Review:**

This paper argues that text generated by existing neural language models are not as good as real text and proposes two metric functions to measure the distributional difference between real text and generated text. The proposed metrics are tried on language GANs but fail to produce any improvement.

Major issues:

This manuscript is poorly organized and the introduction is not well-written. It’s true that generating text from random noise vector remains a challenging problem, but sequence-to-sequence models for machine translation and question answering have achieved tremendous successes. The description in the first paragraph about neural language models is not accurate.

There are numerous grammar issues and mis-spellings. For e.g., pp. 1: “RelGAN which needs not...”, pp. 2: “We analysis…”, “could be find…”, pp 3: “equation 8” should be “equation 9”...

The proposed metrics are also questionable. Eq. 3 on page 2 holds for any x sampled from the distribution, not just for a single data point. To test the effectiveness of a good metric, extensive experiments on toy datasets such as MNIST, CIFAR10, and synthetic datasets should be conducted. This paper mixes text generation and proposed metrics together. The claimed failure experiments make the proposed metrics even more questionable.

In summary, the presentation and the organization of this paper should be significantly improved for submission. The proposed metrics are questionable and should be thoroughly tested on synthetic and toy datasets before deploying it for text generation.

**Experience Assessment:**

I have published one or two papers in this area.

**Review Assessment: Checking Correctness Of Derivations And Theory:**

I assessed the sensibility of the derivations and theory.

**Review Assessment: Checking Correctness Of Experiments:**

I assessed the sensibility of the experiments.

**Review Assessment: Thoroughness In Paper Reading:**

I read the paper thoroughly.

---

### Decision · Program_Chairs · 2019-12-19

**Decision:**

Reject

**Comment:**

The authors propose a novel metric to detect distributional discrepancy for text generation models and argue that these can be used to explain the failure of GANs for language generation tasks. The reviewers found significant deficiencies with the paper, including:

1) Numerous grammatical errors and typos, that make it difficult to read the paper.

2) Mischarcterization of prior work on neural language models, and failure to compare with standard distributional discrepancy measures studied in prior work (KL, total variation, Wasserstein etc.). Further, the necessity of the complicated procedure derived by the authors is not well-justified.

3) Failure to run experiments on standard banchmarks for image generation (which are much better studied applications of GANs) and confirm the superiority of the proposed metrics relative to standard baselines.

The reviewers were agreed on the rejection decision and the authors did not participate in the rebuttal phase.

I therefore recommend rejection.